# Optimization of Ultrasonic-Assisted Extraction of Active Components and Antioxidant Activity from *Polygala tenuifolia*: A Comparative Study of the Response Surface Methodology and Least Squares Support Vector Machine

**DOI:** 10.3390/molecules27103069

**Published:** 2022-05-10

**Authors:** Xuran Li, Simiao Chen, Jinghui Zhang, Li Yu, Weiyan Chen, Yuyan Zhang

**Affiliations:** Department of Life, Zhejiang Chinese Medical University, Hangzhou 31000, China; lixuran1998@163.com (X.L.); simiaochen1@outlook.com (S.C.); zhangjinghui@sina.com (J.Z.); yuli9119@126.com (L.Y.); chenweiyan19@sina.com (W.C.)

**Keywords:** Yuanzhi, 3,6′-disinapoylsucrose, polygalaxanthone III, response surface methodology, least squares support vector machine, antioxidant activity

## Abstract

Dried roots of *Polygala tenuifolia* (YuanZhi in Chinese) are widely used in Chinese herbal medicine. These components in YuanZhi have significant anti-oxidation properties owing to high levels of 3,6’-disinapoylsucrose (DISS) and Polygalaxanthone III (PolyIII). In order to efficiently extract natural medicines, response surface methodology (RSM) and least squares support vector machine (LSSVM) were used for the modeling and optimization of ultrasound-assisted extraction of DISS and PolyIII together to determine the antioxidant activity of the extracts obtained from YuanZhi. For the optimal combination of the comprehensive yield of DISS and PolyIII (Y), the Box-Behnken design (BBD) was used to improve extraction time (X_1_), extraction temperature (X_2_), liquid–solid ratio (X_3_), and ethanol concentration (X_4_). The optimal process parameters were determined to be as follows: extraction time, 93 min; liquid–solid ratio, 40 mL/g; extraction temperature, 48 °C; and ethanol concentration, 67%. With these conditions, the predictive optimal combination comprehensive evaluation value is 13.0217. It was clear that the LS-SVM model had higher accuracy in predictive and optimization capabilities, with higher antioxidant activity and lower relative deviations values, than did RSM. Hence, the LS-SVM model proved to be more effective for the analysis and improvement of the extraction process.

## 1. Introduction

*Polygala tenuifolia* (Yuanzhi) is widely distributed in China, and its roots are an important herb used in traditional Chinese medicine and have been widely reported to have multiple physiological roles and to produce a variety of biological effects, such as antioxidant, analgesic and anti-inflammatory activities. More recently, there has been renewed interest in treating neurocognitive system diseases of Yuanzhi root extracts. In addition, Yuanzhi is the main component of “Kai Xin Powder”, a commonly used classic recipe [1]. At present, it has been proven that oligosaccharide ester compounds and xanthones are the major bioactivity constituents of Yuanzhi [2,3]. Noticeably, 3,6’-disinapoylsucrose (DISS) and Polygalaxanthone III(PolyIII) are the main ingredients in Yuanzhi, and they have significant antioxidant activity [4]. This means that DISS and PolyIII have good scavenging ability to active oxygen free radicals [5,6]. DISS has protective effects on brain and neuron cells and has antidepressant and anxiolytic effects [7,8]. PolyIII has an excitatory effect in the central nervous system and inhibition of monoamine oxidase [9] and has antidepressant and anti-inflammatory effects [10]. Meanwhile, these compounds constitute the quality control indices of Yuanzhi in the Chinese pharmacopeia. In conclusion, Yuanzhi can be used to treat disorders such as depression and AD [11], each dose in 3–10 g from *Chinese pharmacopeia 2020* [12]. In addition, DISS and PolyIII can be used as high value natural antioxidant products. In order to develop traditional Chinese medicine, it has important scientific significance to make better use of resources and improve the yield of DISS and PolyIII (Figure 1).

In general, the traditional extraction methods such as percolation, decoction, impregnation and reflux consume large amounts of solvents and energy. Compared with traditional extraction methods, ultrasound-assisted extraction (UAE) can utilize the mechanical, cavitation and thermal effects of ultrasonic waves to overcome the weaknesses of traditional extraction and significantly improve the extraction efficiency with lower consumption [13,14,15]. Exactly, the propagation of ultrasonic mechanical thermal vibration contributes to the dissolution and diffusion of plant active ingredients [16]. The cavitation releases energy and generates instantaneous pressure and high temperature to cause the cell wall to be cracked, which prompts the contents that are directly dissolved and fully contacted with the solvent [17]. The process is not associated with damage to phytochemicals; it has been widely used to extract biologically active components from plants [18]. In addition, the solvent extraction process is affected by many factors, particularly extraction factors including solvent concentration, extraction time, extraction temperature and liquid–solid ratio, and these factors can individually or together affect extraction efficiency. Because of multitudinous factors, it is a time-consuming and expensive process to analyze and optimize the extraction condition without a mathematical model. Therefore, it is essential to use multivariate analysis methodologies, such as response surface methodology (RSM) or least squares support vector machine (LSSVM) to analyze the interactive effects of these factors and optimize extraction parameters [19].

RSM is an optimization method for comprehensive experimental design and mathematical modeling proposed by Box and Wilson in 1951. It is commonly used to understand the performance of complex systems and to optimize any type of complex extraction process [20,21]. RSM, which has been widely applied in the extraction from plants or herbs due to its advantages of fewer trials, higher precision, better predictive performance, and an optimal response, can be obtained through the visual inspection of the three-dimensional response surface, can elucidate the interactive effect of factors on target responses, and can optimize chemical and physical processes. Compared with the RSM, least squares support vector machine (LSSVM) is derived from the support vector machine (SVM) approach and is a further improvement of the support vector machine model [22], which improves the operation speed by transforming the objective function, optimizing the equation conditions, and reducing computational complexity. It has been widely used in many fields [23,24]. A comparative study of RSM and LSSVM can provide stronger evidence while comparing methods in analytical chemistry. To the best of our knowledge, a comparison between RSM and LSSVM models has not been reported with regard to the simultaneous optimization of comprehensive yields of DISS and PolyIII and the antioxidant activity of Yuanzhi. This study is the first attempt to optimize UAE conditions for DISS and PolyIII and their antioxidant activities (2,2-diphenyl-1-picrylhydrazyl radical scavenging capacity (DPPH) and 2, 2’-azino-bis (3-ethylbenzothiazoline-6-sulfonic acid) antioxidant power (ABTS)) by Box–Behnken design (BBD). Furthermore, extraction time (X_1_), extraction temperature (X_2_), liquid–solid ratio (X_3_), and ethanol concentration (X_4_) were selected as independent variables in order to optimize the extraction process for yield and antioxidant activity with four-factor three-level BBD.

## 2. Results

### 2.1. Single-Factor Experiments

#### 2.1.1. Effect of Extraction Time

The extraction times were set at 30, 60, 90, 120, and 150 min. Other experimental conditions were as follows: extraction temperature was 40 °C, liquid–solid ratio was 8 mL/g, and 70% ethanol was selected as solvent. The effect of extraction time on response value is shown in Figure 2a. With the prolongation of extraction time, the response value increased. After the extraction time lasted for 90 min, the response value reached its maximum, and was followed by a declining trend, which might be due to the degradation of DISS and PolyIII or the increase in the dissolution of other impurities.

#### 2.1.2. Effect of Extraction Temperature

Extraction temperature is an important factor during the extraction process [25]. The experiments in this study were conducted at 20, 30, 40, 50, 60 °C to investigate the influence of extraction temperature. The extraction time was set to 90 min, the liquid–solid ratio was 10 mL/g, and the ethanol concentration was 70%. All other conditions were equal. As shown in Figure 2b, the study results demonstrated that the response value increases gradually with the increase in temperature, reaching the maximum value at 50 °C, and then decreasing slowly. Therefore, the ultrasonic extraction temperature was set to 40, 50 and 60 °C.

#### 2.1.3. Effect of Liquid–solid Ratio

The extraction procedure was executed at different liquid–solid ratios (6, 8, 10, 12, 14 mL/g), while other extraction parameters were as follows: extraction temperature was 50 °C, ethanol concentration was 70%, and extraction time was 90 min. As shown in Figure 2c, the response value increased with the liquid–solid ratio, while the liquid–solid ratio reached 12 mL/g, increasing slowly. Considering cost savings, the extraction liquid–solid ratio was selected as 10, 12, and 14 mL/g.

#### 2.1.4. Effect of Ethanol Concentration

In this study, we investigated the effect of different ethanol concentrations (40%, 55%, 70%, 85%, 95%) on the extraction of DISS and PolyIII. As shown in Figure 2d, the variation trend of the response value first increased and then decreased with the increase in ethanol content. This may be due to increased dissolution of other alcohol-soluble substances, and this may also be that the viscosity of the solvent becomes larger due to the excessive concentration of ethanol, which is not conducive to the dissolution of the compound.

Finally, three levels were determined as follows: the extraction time was 60, 90 and 120 min; liquid–solid ratio was 8, 10 and 12 mL/g; the ethanol concentration were 50%, 70%, 85%, as shown in Table 1.

### 2.2. BBD Method Optimization of Extraction Conditions

The ranges for the four independent extraction variables, namely, extraction time (X_1_), extraction temperature (X_2_), liquid–solid ratio (X_3_) and ethanol content (X_4_), were at three levels (+1, 0, −1) for extraction parameter optimization by the BBD method.

A comprehensive evaluation value of DISS and PolyIII extraction yield was chosen as the dependent variable. In the extraction optimization experiment, 30 groups (5 g for Yuanzhi, *n* = 3) were selected, and the BBD matrix of 30 experimental standard runs was designed by the software, as shown in Table 2.

### 2.3. Model Fitting

#### 2.3.1. RSM Modeling

The optimization results of ultrasound-assisted extraction of Yuanzhi with RSM and the data obtained from the BBD were analyzed to obtain the quadratic polynomial model by regression analysis. The quadratic polynomial model is as follows:(1)$$Y=\beta_{0}+\overset{4}{\underset{z=1}{\sum }}\beta_{i}x_{i}+\overset{4}{\underset{i=1}{\sum }}\overset{4}{\underset{j\geq 1}{\sum }}\beta_{ij}x_{i}x_{j}+\overset{4}{\underset{z=1}{\sum }}\beta_{ii}x_{i}^{2}+\varepsilon$$
where Y represents the comprehensive evaluation value, *X_i_* or *X_j_* (*i*, *j* = 1, 2, 3, 4) are the independent variables: X_1_, extraction time; X_2_, extraction temperature; X_3_, solid–liquid ratio; X_4_, ethanol concentration. *β*_0_ is the constant coefficient. Βi, *β_ij_*, *β_ii_* are constant regression coefficients of the model. *ε* is the residual between model and experiment.

Based on BBD, 30 experiments of the extraction were carried out to study the effect of these four factors on the yields of DISS and PolyIII. The results of the four independent variables and their levels, as well as the experimental and predictive dependent variables based on the selected BBD, are shown in Table 2. From the results, the comprehensive evaluation value of DISS and PolyIII extraction yield in Yuanzhi extracts ranged from 11.4212 to 13.1058. The extraction yield value of comprehensive evaluation value can be expressed as the following equation:Y = 13.01 + 0.17X_1_ + 0.055X_2_ + 0.11X_3_ − 0.28X_4_ − 0.083X_1_X_2_ + 0.045X_1_X_3_ − 0.095X_1_X_4_ − 0.055X_2_X_3_ − 0.22 X_2_X_4_ + 0.058X_3_X_4_- 0.33X_1_^2^ − 0.32X_2_^2−^ 0.13X_3_^2^ − 0.86X_4_^2^(2)

The analysis of variance (ANOVA) is an effective and important method to evaluate the quality and significance of mathematical models; the results are shown in Table 3. *p* values were used to check the significance of each coefficient. When the “*p*-value > f” is less than 0.05, 0.01 and 0.001, it indicates that the model item is significant, highly significant and extremely significant, respectively. When the *p* value < 0.05 (α = 0.05), the fitted regression equation was considered statistically significant.

On the basis of the ANOVA results, they showed significant linear (X_1_, X_2_, X_3_ and X_4_) and interactive (X_1_X_2_, X_1_X_4_, X_2_X_4_) effects on comprehensive evaluation values. The coefficient of determination value (R^2^) is in reasonable agreement with the adjusted determination coefficients (Adj. R^2^), which means that the difference is less. The lack of a fit F-value of 0.82 implies that the lack of fit is not significant relative to the pure error. There is a 63.19% chance that a lack of fit F-value this large could occur due to noise. In this study, the weight transformation method was used for the optimization of the DISS and PolyIII. Based on the regression equations, 3D and 2D response surface plots were established to explain the individual as well as the interactive influences of the four factors on the comprehensive evaluation value of DISS and PolyIII yields.

The response surface 3D plots and 2D contour plots (Figure 3) were designed to show the interaction effects of the independent variables on the yields of the dependent variables. The higher the slope of the 3D graph is, the steeper the slope, indicating a more significant interaction between the two factors and a greater the influence of the two factors on the response value. The gentler the 3D graph is, the less obvious the interaction between the two factors, and the less influence on the response value. The more circular the 2D graph is, the smaller the interaction between the two factors; the more elliptical the 2D graph is, the greater the interaction between the two factors. Above all, the F test, *X*_1_, *X*_3_, *X*_4_, *X*_2_*X*_4_, *X*_1_^2^, *X*_2_^2^, *X*_4_^2^ were extremely significant parameters, *X*_3_^2^ was a highly significant parameter, *and X*_2_*, X*_1_*X*_2_, *X*_1_*X*_4_ were significant parameters. Combined with Figure 3, it can be seen that extraction time and ethanol concentration had a greater effect on extraction efficiency.

Based on the Box–Behnken design and the response surface plots, the optimal comprehensive evaluation value is 13.0870, and the result from the verification experiment was 12.8650. The suggested conditions were: extraction time 99.06 min, extraction temperature 50.70 °C, liquid–solid ratio 10.88 mL/g, ethanol concentration 67.39%. For the convenience of experiments, parameters were modified slightly in the verification experiment as follows: extraction time 99 min, extraction temperature 51 °C, liquid–solid ratio 11 mL/g, ethanol concentration 67%.

#### 2.3.2. LS-SVM Modeling

The values of kernel parameter *g* and penalty factor *C* are closely related to the data sample set and affect the learning and generalization ability of the model itself for the LS-SVM model. The higher the penalty factor *C* value is, the smaller the allowable error. However, too much *C* value will lead to too much fitting and the generalization ability will be reduced. A smaller *C* value can enhance the generalization ability of the LS-SVM model, and the allowable error will also be larger. If the input range of the experimental data samples is large, the value of the core parameter g needs to be increased. Otherwise, the value of the kernel parameter g needs to be reduced.

For the small amount of data in this experiment, the radial basis function (RBF) has a high performance and application range. Therefore, when the RBF kernel function is selected, an appropriate kernel parameter *g* and penalty factor *C* should be selected simultaneously to optimize the LS-SVM model. In this study, 30 groups of experimental data of BBD analysis scheme are preprocessed. The experimental data matrix of 30 rows and 5 columns is divided into an input variable matrix of 30 rows and 4 columns and an output variable matrix of 30 rows and 1 column into the platform of Matlab 2018b software (MathWorks, Natick, MA, USA). The output variables of the same input variables are averaged before the input. There are only 25 rows and 4 columns of the input variable matrix and 25 rows and 1 column of the output variable. LS-SVM was used for modeling, and the kernel function was RBF function. Finally, the cross verification method is used to verify the nuclear parameter *g* and penalty factor *C* simultaneously. The effect of each pair of parameters is tested one by one in the parameter matrix composed of *g* and *C*, and the optimal kernel parameter *g* = 0.1 and the optimal penalty factor C = 10.4 are obtained. The process is shown in Figure 4.

By analyzing the results of the optimal parameters in Figure 4 and combining them with the program results, the optimal core parameter G = 0.1 and penalty factor C = 10.4 can be obtained. After obtaining appropriate nuclear parameter G and penalty factor C, the comprehensive evaluation predicted values of 25 groups of BBD analysis schemes can be obtained by returning to the LS-SVM model again, as shown in Table 4.

The mean-square error (*MSE*) was used to evaluate the performance of the LS-SVM model, and its specific formula is as follows.
(3)$$MSE=\frac{1}{n}\sum_{i=1}^{n}{\left(y_{i}-y_{i}^{*}\right)}^{2}$$
where *n* is 25 data sets, $y_{i}$ is LS-SVM predicted value, $y_{i}^{*}$ is the actual value obtained by the experiment.

Finally, 25 groups of original experimental values and predicted values were substituted into the formula to calculate *MSE*, and a comparison chart was drawn between the real comprehensive evaluation value and predicted comprehensive evaluation value of the experiment, as shown in Figure 5.

Figure 5 shows that the comprehensive evaluation values of 25 groups of actual data and predicted data are close, indicating that the training data fit is close to the actual test data, and the calculated *MSE* is 0.1357, indicating that the model training and prediction effect is good.

Based on the BBD experimental design, combined with the actual experimental reachable conditions, the software Matlab 2018b was used to add 80 groups of gradient data sets (gradient 0.05) for the four factors, and the LS-SVM model was used to predict the optimal combination. Finally, the optimal extraction conditions were as follows: temperature 48 °C, time 93 min, liquid–solid ratio 10 mL/g and ethanol content 73%. The comprehensive evaluation value of the optimal extraction conditions was 13.0217.

### 2.4. Validation Experiment and Comparison between RSM and LSSVM

In this study, RSM and LS-SVM models are used to model and optimize the data, and both models can fit the experimental data well. All experiments under the optimized conditions by RSM and LS-SVM were performed in quintuplicate, and the results are displayed in Table 5. The results of comparison of antioxidant activity are shown in Figure 6 (the concentrations were 0.2, 0.4, 0.6, 0.8, 1.0 mg/mL).

It can be seen from Table 5 that these findings indicate that the performance of LS-SVM was better than RSM. Not only that, LS-SVM had larger response values than those of RSM under the predicted optimum conditions. From Figure 6, the antioxidant activity of extraction according to the LS-SVM model was better than the RSM model. In conclusion, LS-SVM has better prediction ability and optimization ability than RSM.

### 2.5. HPLC Analysis of DISS and PolyIII

The calibration curve of DISS was Y = 13.965x + 0.553, R² = 0.9995, showing a good linear relationship in the concentration range of 1.2–2 mg/mL. The calibration curve of PolyIII was 22.106x − 2.521, R² = 0.9991, showing a good linear relationship in the concentration range of 0.1–0.5 mg/mL. The HPLC profiles of some standard substances and samples are shown in Figure 7.

DISS and PolyIII content were measured by the corresponding peak area transformation. The results showed that the content of DISS was higher than that of PolyIII in the sample. Under these conditions, the retention times of DISS and PolyIII were 8.73 and 25.4 min, respectively.

The RSD values of peak area for DISS and PolyIII were 0.85% and 0.56% of stability, which indicated that it was great from 0 to 16 h at room temperature. The repeatability and precision of DISS and PolyIII were 0.77%, 0.69% and 0.53%, 0.81%, respectively.

## 3. Discussion

Above all, when the extraction time, extraction temperature, liquid–solid ratio and ethanol concentration were increased, the comprehensive evaluation value increased at the beginning and then decreased. The cavitation effect of ultrasound facilitated the extraction of DISS and PolyIII from the herb. Nevertheless, prolonged ultrasound may cause degradation of chemical structures and may reduce the extraction efficiency of DISS and PolyIII [26]. The increase in temperature strengthened the diffusion and dissolution of solute in the solvent and decreased the viscosity of the solvent itself, thus increasing the diffusion rate of the solvent. However, with the temperature continuously increased, the cavitation effect became weak and the yield reduced [27], which might be due to the reduction of solvent surface tension at high temperature; thus, the effective components were decreased in the dissolution [28]. The yield of DISS and PolyIII increased with the liquid–solid ratio in a certain range and then decreased slowly after the peak. This was due to the viscosity of the solution being high in the beginning, where it was difficult to produce a cavitation effect. With the increase in the liquid–solid ratio, the viscosity and concentration of the solvent decreased, which led to a greater cavitation effect. As cavitation becomes stronger, the extraction efficiency of effective substances also increases. However, excessive cavitation may lead to the destruction of the active substance and degradation of the solvent itself [29,30]. As for the ethanol concentration, after the peak concentration, the extraction yield decreased with increasing concentration. This might have been due to the polarity of DISS and PolyIII. It could also be due to the increase in ethanol concentration, in which the dielectric constant of the solvent decreases and the solubility and diffusivity of phenolic compounds increase. Therefore, the dissolution of the other substances is reduced.

RSM is a widely used method for TCM extraction [31,32,33], including statistical factorial experimental design, modeling between causal factors and response variables, and multi-objective optimization to seek optimal formulation. The theoretical model is composed of multiple formula factors and process variables, and the number of tests for model preparation can be greatly reduced by adopting the composite experimental design. Through the combination of causal factors, the response variables of each model formula are predicted quantitatively. The traditional method is to apply multiple regression analyses based on quadratic polynomial equations [34]. Finally, a multi-objective optimization algorithm is used to predict the optimal formula, but the prediction based on the quadratic polynomial is usually limited to a low level because the theoretical relationship between causal factors and the response variables is not clear. This can lead to an underestimation of the optimal formulations [35].

The least square support vector machine (LSSVM) is based on the support vector machine (SVM) algorithm and is an extended version of SVM. It can be used to predict the relationship between multiple factors and their interactions with evaluation indexes [36]. LSSVM is characterized by solving a set of linear equations, instead of the traditional more complex quadratic programming method to avoid an insensitive loss function; meanwhile, it greatly reduces the complexity of calculation, speeds up the solution, and expands the application field [37], which has been used in fault diagnosis [38], electricity consumption [39], water quality prediction [40] and other fields. However, it is seldom used in pharmacology, especially in the extraction process of effective substances of TCM. There are many exploitable capabilities in LSSVM that have yet to be explored by researchers.

Compared with RSM and other machine learning regression methods, such as random decision forest [41] and the artificial neural network (ANN) model [42,43,44], LSSVM has certain advantages, which will provide a new idea and reference for the study of extraction process optimization of TCM active ingredients.

In this study, the simultaneous optimization of the comprehensive evaluation values of DISS and PolyIII, RSM and LS-SVM modeling methods were used to compare the antioxidant activity of Yuanzhi extracts. The results indicate that the extracts under optimal conditions of the LS-SVM model have a better yield of extraction and better antioxidant activity than the RSM model extracts. Therefore, the LS-SVM model has good predictive ability, and the results are satisfactory. The optimal predictive conditions obtained using LS-SVM are as follows: extraction time 93 min; liquid–solid ratio 40 mL/g; extraction temperature 48 °C; and ethanol concentration 67%. With these conditions, the predictive optimal combination comprehensive evaluation value is 13.0217. It can be said that RSM is a widely used way to optimize extraction processes [45]. LS-SVM can be used as a potential alternative technique. In the future, the findings of this study can provide effective guidance for the extraction process of natural antioxidants DISS and PolyIII and can be used for industrial large-scale extraction.

## 4. Materials and Methods

### 4.1. Materials

The dried roots of *Polygala tenuifolia* (batch no. 210413) were purchased from the Chinese Medicine Pieces Co., Ltd. (Hangzhou, China) of Zhejiang Chinese Medical University and were identified by Professor Yuyan Zhang of the Zhejiang Chinese Medical University. 3,6’-disinapoylsucrose reference substance (B21780-20 mg) and Polygalaxanthone III reference substance (B21625-20 mg) were purchased from Shanghai Source Leaf Technology Co., LTD. (Shanghai, China). Microporous membrane (0.22 micron) was purchased from Jingteng Co., LTD (Tianjin, China). HPLC-grade acetonitrile was purchased from Merck (Darmstadt, Germany). Deionized water was prepared using a Millipore water purification system (Millipore Co., Ltd., Billerica, MA, USA). Ethanol (the mass fraction of ethanol ≥99.7%, CAS:64-17-5) was purchased from SHUANGLIN chemical reagent Co., LTD. (Hangzhou, China). Unless stated otherwise, all reagents and chemicals were of analytic grade.

### 4.2. Experimental Design

#### 4.2.1. Single-Factor Experiment

According to the large number of literature reviews and summaries, the extraction time (30, 60, 90, 120,150 min), extraction temperatures (20, 30, 40, 50, 60 °C), liquid–solid ratio (6, 8, 10, 12, 14 mL/g), and ethanol concentrations (40, 55, 70, 85, 95%) were selected for single-factor experiments

#### 4.2.2. Variables Selection and Weight Design of Multi-Component Indexes

In general, the UAE yield of Yuanzhi was influenced by various factors, which included extraction temperature, solvent-to-liquid ratio, ultrasonic power, extraction time, solvent concentration, particle size and so on. Samples sifted through a 65-mesh sieve were subjected to a fixed ultrasound power of 120 W in this study. According to the previous studies, the appropriate solvent is important for the extraction of herbs. Owing to its non-toxic nature and an ability to provide higher solubility or extraction yields, ethanol was used as a solvent for the extraction of DISS and PolyIII. Considering costs and safety parameters, ethanol proved to be an ideal extraction solvent. Subsequently, extraction temperature (°C, A), extraction time (min, B) liquid–solid ratio (mL/g, C) and ethanol concentration (%, D) were selected as variables. The single-factor experiment was carried with these variables. The comparison matrix of evaluation indicators is designed by using the Precedence Chart (shown in Table 1). The Precedence Chart (PC) was first proposed by American P.E. Moody in 1983. The scoring method is as follows: Index A and B, if index A is more important than index B, index A receives 1 point. If the effect of indicator B is greater than that of indicator A, then indicator A receives 0 points. If indicator A and indicator B are equally important, they each receive 0.5 points. According to the quality requirements of the Chinese pharmacopoeia 2020 edition, a = DISS, b = PolyIII, and a > b were determined. After data normalization, the unit limit was removed, the weight of each index was multiplied by the normalization score, and then the sum was added to obtain the final comprehensive score. Comprehensive evaluation value (Y) = extraction yield of DISS×0.75 + extraction yield of PolyIII score×0.25.

#### 4.2.3. Box-Behnken Design (BBD) for Extraction Optimization

The Design-Expert (Minneapolis, MN, USA) software was used for experimental design, RSM model regression analysis, and RSM model optimization. Based on the single-factor experimental results, the Box–Behnken design (BBD) was developed with four variables and three levels. The four independent variables were set at 3 levels (−1, 0, 1), with X_1_ (60, 90, 120 min), X_2_ (40, 50, 60 °C), X_3_ (8, 10, 12 mL/g,), X_4_ (55%, 70%, 85%). Details are shown in Table 1. The comprehensive evaluation value of DISS and PolyIII extraction yield was chosen as the dependent variables. In the extraction optimization experiment, 30 groups (5 g for Yuanzhi, *n* = 3) were selected, and the BBD matrix of 30 experimental standard runs was designed by the software.

#### 4.2.4. Least Squares Support Vector Machine for Extraction

The least squares support vector machine (LSSVM) model can explain several extraction factors (x_1_, x_2_… x_i_) and the comprehensive evaluation index Y of the extraction rate. The basic for the LS-SVM is that the i-th data input X_i_ = (X_1_, X_2_… X_n_) represents the corresponding n extraction factor values, and the i-th data output Yi represents the comprehensive evaluation value of the corresponding extraction rate. Given the known experimental data set, D = was used to find the quantitative relationship between extraction factors and the comprehensive evaluation value. To be specific, the LSSVM was used to establish the regression model of the comprehensive evaluation value on the extracted factors in the high dimensional feature space, and then the Lagrange multiplier was introduced according to the structural risk minimization principle under Kuhn–Tucker conditions (KKT) of optimization theory. The regression model was transformed into an optimization model related to the kernel function. Finally, the prediction comprehensive evaluation value with the smallest error between the real comprehensive evaluation value and the optimal parameters can be found through continuous learning of the data set.

In this study, four extraction process data were set up based on the Matlab language environment, namely, extraction time (X_1_), liquid–solid ratio (X_2_), ultrasonic temperature (X_3_), ethanol content (X_4_) and comprehensive evaluation value Y. In addition, Y is the comprehensive evaluation value of the extraction rate of DISS and PolyIII in Yuanzhi by linear weighted calculation by the precedence chart method.

### 4.3. Ultrasound-Assisted Extraction

The dry powder of Yuanzhi (5 g) was placed in a 250 mL conical flask and soaked in ethanol with a certain concentration and liquid–solid ratio according to the BBD experimental conditions. Then, the conical flask was placed in an ultrasonic cleaner. The UAE model was KQ5200DE, which was purchased from Kunshan Ultrasonic Instrument Co., LTD. The samples were extracted at 40 kHz and 120 W. The extraction time and temperature were selected based on the BBD experimental conditions. After extraction, each extract was filtered with a Buchner funnel, and the filtrate was passed through a 0.22 μm microporous membrane. Lastly, the filtrates were analyzed by HPLC.

### 4.4. Determination of Antioxidant Activity

The antioxidant activity of Yuanzhi extracts was evaluated using the 2,2-diphenyl-1-picrylhydrazyl radical scavenging capacity (DPPH) and 2, 2’-azino-bis (3-ethylbenzothiazoline-6-sulfonic acid) antioxidant power (ABTS) [46]. The DPPH assay was introduced by Brand Williams [47]. The solution of DPPH radicals was prepared by adding 180 µL of 0.1 mM DPPH in anhydrous ethanol and 20 µL of the sample solution at different concentrations (0.2–1.0 mg/mL). The mixture was subsequently incubated for 30 min at room temperature in the dark, and the absorbance of the solution was measured at 517 nm [48]. The DPPH radical scavenging activity was calculated according to the following equation.
(4)$$Scavenging\;percente\left(\%\right)=\left(1-\frac{A2-A0}{A1}\right)\times 100$$
where *A*0 is the absorbance of sample + anhydrous ethanol, *A*1 is the absorbance of anhydrous ethanol + DPPH, and *A*2 is the absorbance of sample + DPPH.

The ABTS solution was adjusted with absolute ethanol to make 0.70 ± 0.20 of the absorbance (734 nm). The reaction mixture contained 20µL sample solution with different concentrations (0.2–1.0 mg/mL) and 180µL ABTS solution. After reacting for 6 min at room temperature, the absorbance was determined at 734 nm.
(5)$$Scavenging\;percente\left(\%\right)=\left(1-\frac{A2-A0}{A1}\right)\times 100$$
where *A*1 is the absorbance of absolute ethanol instead of sample, *A*2 is the absorbance of the sample, and *A*0 is the absorbance of absolute ethanol instead of ABTS.

### 4.5. HPLC Analysis

HPLC analysis was carried out using a Thermo UltiMate 3000 HPLC with VWD-3100 UV detector, equipped with column Kromasil C18 (4.6 × 250 mm, 5μm). The mobile phase consisted of 0.05% phosphoric acid in water (A) and acetonitrile (B). In addition, the flow rate was 1 mL/min. The column temperature was set at 25 °C, and the injection volume was 10 μL. The monitoring wavelength for Yuanzhi was 320 nm [49].

The isocratic elution was as follows: 0–30 min, 16% (B), with a total run time of 30 min. Under such chromatographic conditions, DISS and PolyIII were well separated. Sample peaks were identified based on their retention time compared with the standard. A standard curve to quantify DISS and PolyIII was constructed in the range of 1.2, 1.4, 1.6, 1.8, 2.0 mg/mL and 0.1, 0.2, 0.3, 0.4, 0.5 mg/mL, respectively.

## 5. Conclusions

In this study, the pharmacological value and action of Yuanzhi and its main compounds were described. Then, RSM and LS-SVM were compared, and the optimal extraction process and antioxidant activity were determined by the final DISS and POLY response values. The results showed that the best extraction process of Yuanzhi was at temperature 48 °C, time 93 min, liquid–solid ratio 10 mL/g and ethanol content 73%. The comprehensive evaluation value of the optimal extraction conditions was 13.0217. In addition, the extraction of Yuanzhi had great antioxidant activity. Our experiments showed that the extraction from Yuanzhi by LS-SVM had better yield and antioxidant activity than from RSM. In the future, Yuanzhi will be better developed and utilized as an antidepressant and AD treatment drug. The method in this paper is feasible, with a high extraction rate and good antioxidant activity, which is useful for the application of Yuanzhi in food, medicine and the health product industry.

## Figures and Tables

**Figure 1 molecules-27-03069-f001:**
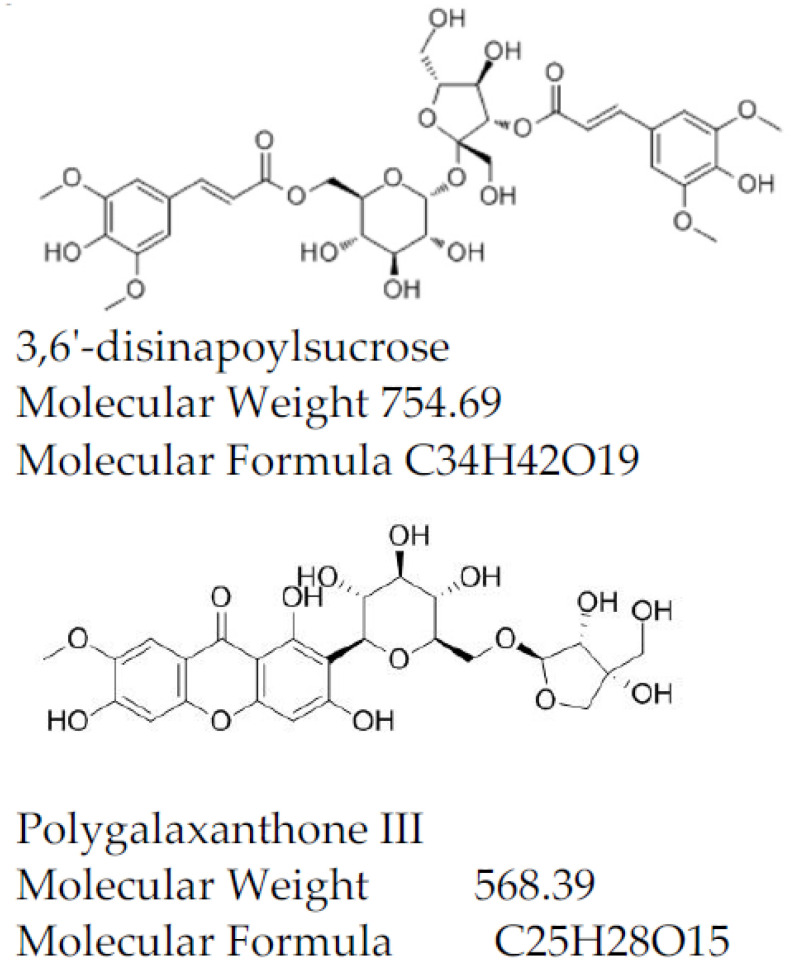
Chemical structure of 3,6’-disinapoylsucrose and Polygalaxanthone III.

**Figure 2 molecules-27-03069-f002:**
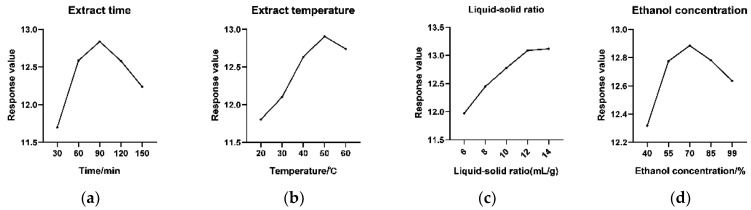
Single-factor experimental results of Yuanzhi extraction process. (**a**) Effect of extract time. (**b**) Effect of extraction temperature. (**c**) Effect of extraction liquid–solid ratio. (**d**) Effect of extraction ethanol concentration.

**Figure 3 molecules-27-03069-f003:**
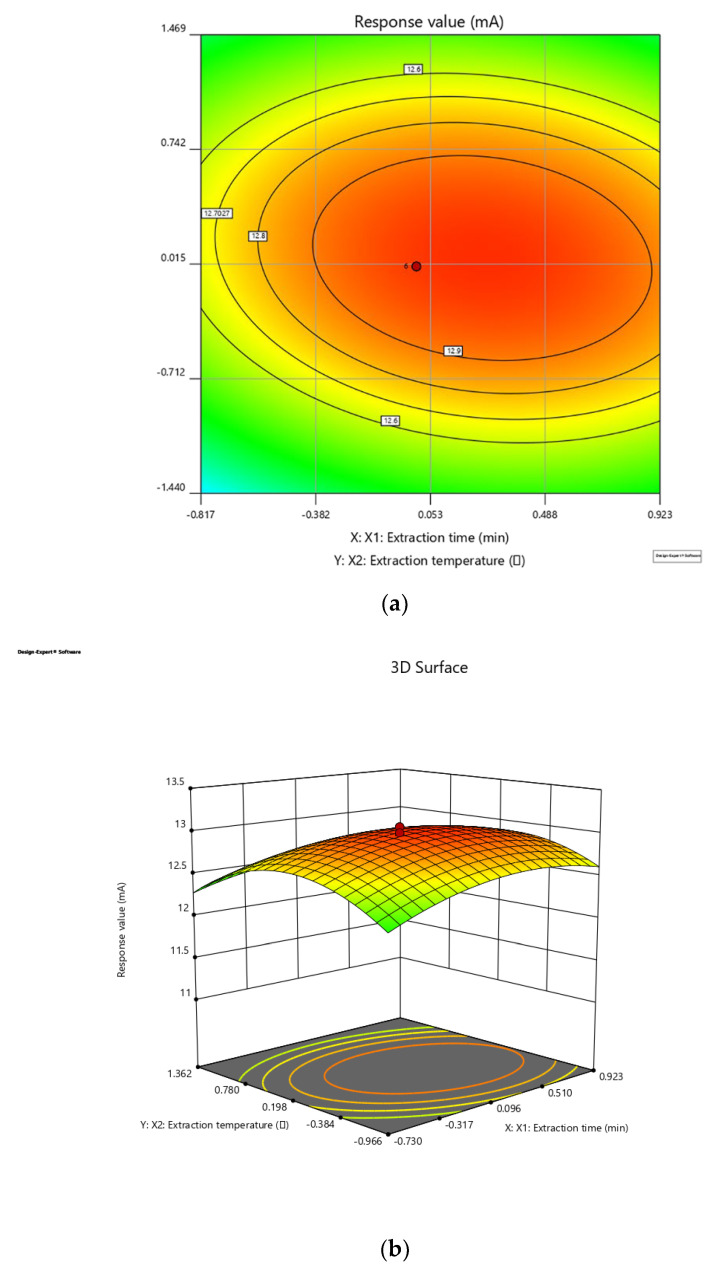
Response surface 3D plots and 2D contour plots. (**a**,**b**) The interaction effects of time and temperature; (**c**,**d**) the interaction effects of time and liquid–solid ratio; (**e**,**f**) the interaction effects of time and ethanol concentration; (**h**,**i**) the interaction effects of liquid–solid ratio and temperature; (**j**,**k**) the interaction effects of liquid–solid ratio and ethanol concentration.

**Figure 4 molecules-27-03069-f004:**
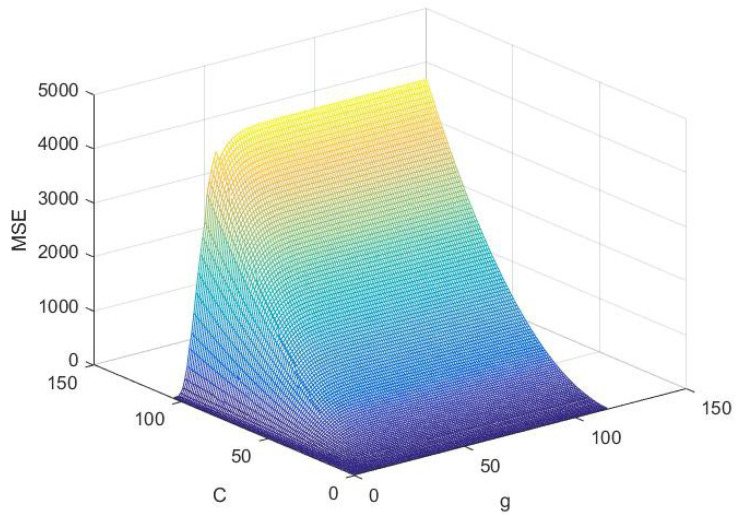
Results of the optimal parameters.

**Figure 5 molecules-27-03069-f005:**
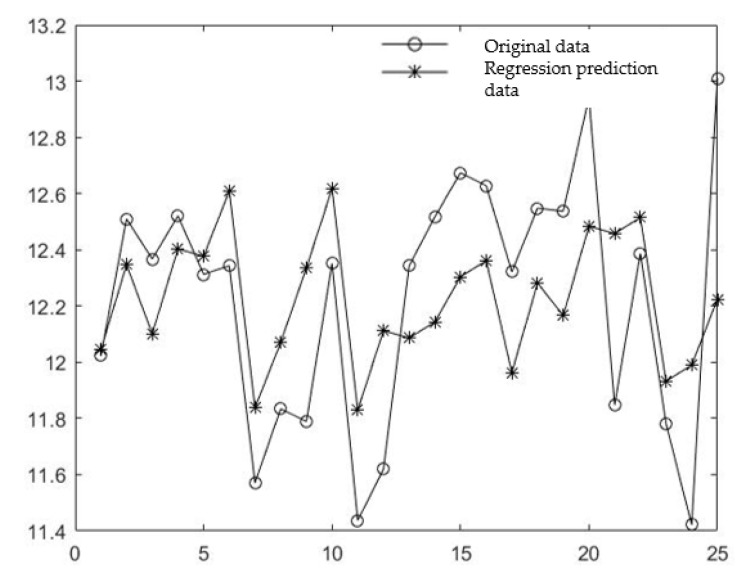
Comparison of experimental data and model prediction data.

**Figure 6 molecules-27-03069-f006:**
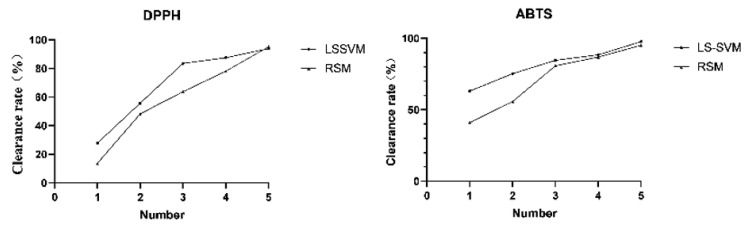
Antioxidant activity of Yuanzhi extracts.

**Figure 7 molecules-27-03069-f007:**
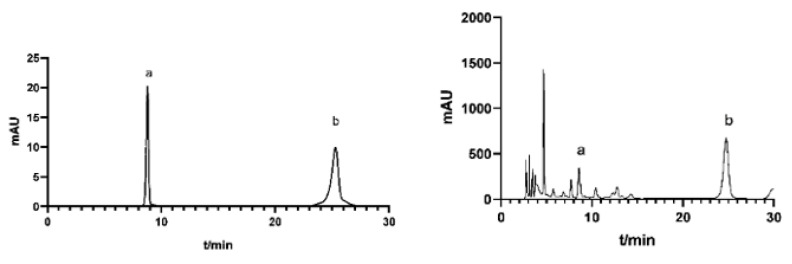
HPLC diagram of Yuanzhi reference substance and Yuanzhi extract (a: PolyIII; b: DISS).

**Table 1 molecules-27-03069-t001:** Factors and levels in Box–Behnken design.

Factor	Symbols	Level		
		−1	0	+1
Extraction time, min	X_1_	60	90	120
Extraction temperature, °C	X_2_	40	50	60
Liquid–solid ratio, mL/g	X_3_	8	10	12
Ethanol concentration, %	X_4_	55	70	85

**Table 2 molecules-27-03069-t002:** Box-Behnken design (BBD) and results.

Std	X_1_	X_2_	X_3_	X_4_	PolyIII (mg/g)	DISS (mg/g)	Y (Synthesis Score)
1	−1	−1	0	0	1.4456	15.5484	12.0227
2	1	−1	0	0	1.5295	16.1695	12.5095
3	−1	1	0	0	1.4981	15.9899	12.3670
4	1	1	0	0	1.5736	16.1695	12.5205
5	0	0	−1	−1	1.4016	15.9492	12.3123
6	0	0	1	−1	1.5044	15.9561	12.3432
7	0	0	−1	1	1.4409	14.9479	11.5711
8	0	0	1	1	1.5486	15.2625	11.8340
9	−1	0	0	−1	1.3442	15.2710	11.7893
10	1	0	0	−1	1.4611	15.9812	12.3512
11	−1	0	0	1	1.3649	14.7946	11.4372
12	1	0	0	1	1.5000	14.9916	11.6187
13	0	−1	−1	0	1.4956	15.9583	12.3426
14	0	1	−1	0	1.5200	16.1822	12.5166
15	0	−1	1	0	1.5716	16.3750	12.6741
16	0	1	1	0	1.6158	16.2980	12.6275
17	−1	0	−1	0	1.3782	15.9690	12.3213
18	1	0	−1	0	1.3829	16.2697	12.5480
19	−1	0	1	0	1.4019	16.2483	12.5367
20	1	0	1	0	1.5831	16.7280	12.9418
21	0	−1	0	−1	1.4724	15.3066	11.8481
22	0	1	0	−1	1.5987	15.9826	12.3866
23	0	−1	0	1	1.5282	15.2000	11.7821
24	0	1	0	1	1.5159	14.7230	11.4212
25	0	0	0	0	1.6715	16.6493	12.9048
26	0	0	0	0	1.7005	16.7102	12.9578
27	0	0	0	0	1.6571	16.9221	13.1058
28	0	0	0	0	1.6694	16.8204	13.0327
29	0	0	0	0	1.6842	16.8039	13.0240
30	0	0	0	0	1.7018	16.8183	13.0392

**Table 3 molecules-27-03069-t003:** Results of analysis of variance (ANOVA) of the Box–Behnken design (BBD).

Source of Variation	Sum of Squares	Variance	Mean Square	F	*p*	Significance
Model	7.3700	14	0.5264	121.9277	< 0.0001	***
*X* _1_	0.3385	1	0.3385	78.4086	< 0.0001	***
*X* _2_	0.0363	1	0.0363	8.4172	0.011	*
*X* _3_	0.1508	1	0.1508	34.9318	< 0.0001	***
*X* _4_	0.9443	1	0.9443	218.7243	< 0.0001	***
*X* _1_ *X* _2_	0.0278	1	0.0278	6.4295	0.0228	*
*X* _1_ *X* _3_	0.0080	1	0.0080	1.8423	0.1947	
*X* _1_ *X* _4_	0.0362	1	0.0362	8.3790	0.0111	*
*X* _2_ *X* _3_	0.0122	1	0.0122	2.8221	0.1137	
*X* _2_ *X* _4_	0.2022	1	0.2022	46.8362	< 0.0001	***
*X* _3_ *X* _4_	0.0135	1	0.0135	3.1175	0.0978	
*X* _1_ ^2^	0.7355	1	0.7355	170.3606	< 0.0001	***
*X* _2_ ^2^	0.7052	1	0.7052	163.3234	< 0.0001	***
*X* _3_ ^2^	0.1103	1	0.1103	25.5470	0.0001	**
*X* _4_ ^2^	5.0848	1	5.0848	1177.7096	< 0.0001	***
Residual	0.0648	15	0.0043			
Lack of fit	0.0402	10	0.0040	0.8199	0.6322	
R^2^	0.9913					
Adjusted R^2^	0.9832					
Predicted R^2^	0.9641					

* Represents significant at *p* < 0.05; ** Represents highly significant at *p* < 0.01; *** Represents extremely significant at *p* < 0.001.

**Table 4 molecules-27-03069-t004:** Predictive values of comprehensive evaluation of the LS-SVM model.

Group	Predicted Value	Group	Predicted Value	Group	Predicted Value
1	12.0440	11	11.8291	21	12.4576
2	12.3464	12	12.1121	22	12.5146
3	12.1006	13	12.0862	23	11.9325
4	12.4031	14	12.1429	24	11.9895
5	12.3770	15	12.3042	25	12.2235
6	12.6092	16	12.3609		
7	11.8379	17	11.9634		
8	12.0701	18	12.2814		
9	12.3350	19	12.1657		
10	12.6180	20	12.4837		

**Table 5 molecules-27-03069-t005:** Verification of experiment results.

	Runs	PolyIII (mg/g)	DISS (mg/g)	Y	Mean ± SD	Predicted Value	Relative Deviation (%)
RSM	1	1.7148	16.6133	12.8887	12.8402 ± 0.0963	13.0870	1.89
	2	1.7233	16.6408	12.9114			
	3	1.6977	16.5620	12.8459			
	4	1.7058	16.6342	12.9021			
	5	1.6924	16.3064	12.6529			
LS-SVM	1	1.6963	16.7761	13.0062	13.0045 ± 0.0405	13.0217	0.13
	2	1.6883	16.6806	12.9325			
	3	1.7103	16.8408	13.0582			
	4	1.7132	16.7804	13.0136			
	5	1.7005	16.7825	13.0120			

## Data Availability

Not applicable.

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
