# Peer review of "Optimization of Ultrasonic-Assisted Extraction of Active Components and Antioxidant Activity from Polygala tenuifolia: A Comparative Study of the Response Surface Methodology and Least Squares Support Vector Machine"

_molecules, 2022, doi:10.3390/molecules27103069_

Round 1

Reviewer 1 Report

Present research by Li et al. is focused on the extraction of bioactives from Polygala tenuifolia and process optimization. Study was performed in order to maximize content of 3,6'-disinapoylsucrose(DISS) and Polygalaxanthone III (PolyIII) as well as to improve antioxidant activity of extracts. Finally, process was optimized by response surface methodology (RSM) and least squares support vector machine.

I have several questions and remarks which have to be answered prior the further consideration, however, my opinion that paper should be rejected due to following observations:

  • Line 312: “The single-factor experiment was carried with these variables” – Where are the results of single factor experiments? How was that performed?
  • Table 1: How did you select the experimental domain for independent variables?
  • Section 4.3. Information about the UAE device is missing.
  • HPLC method should be explained briefly.
  • Lines 114-130: Why is this give in Results?
  • Experiments were performed in Box-Behnken experimental design which is suitable for RSM. How is that compatible with least squares support vector machine?
  • You compared RSM and least squares support vector machine in terms of optimization. Experiments should be performed at optimal conditions again to check and validate results.
  • Comparison with literature is also missing. RSM has been compared with other optimization tools in numerous studies recently. It should have that in mind for improvement of discussion.
  • What about comparison on other levels? What is more suitable for model fitting? What is more suitable for influence analysis? Authors should look beyond the optimization to improve the scientific impact of the paper.
  • Discussion is rather modest and should be reassessed.
  • Conclusions section is utterly problematic. Authors should put some serious effort and write the appropriate conclusions.
  • Title of the paper is not suitable since antioxidant activity and bioactive compounds were not simultaneously optimized! It should be rewritten.
  • Finally, authors should devote more time in careful paper formatting and English editing by native English speaker.
  • My suggestion for the authors is to rewritte and resubmit the paper.

Author Response

Thank you very much for your letter and advice. We have revised the manuscript, and would like to re-submit it for your consideration. We have addressed the comments raised by the reviewers, and the amendments are highlighted in red in the revised manuscript. Point by point responses to the reviewers’ comments are listed below this letter.

  1. Line 312: “The single-factor experiment was carried with these variables” – Where are the results of single factor experiments? How was that performed?

Answer: I am very sorry that I did not write down the single factor experiment results and process before. I have listed the single factor experimental results and procedures in the article In line 99-144.

  1. Table 1: How did you select the experimental domain for independent variables?

Answer: I chose single-factor variables based on literature, books, and previous experiments. “Preparation Process, composition and antioxidant activity of Polygala tenuifolia extract. doi:10.11937/bfyy.201844222” and so on. It was found that the extraction time and extraction temperature were the factors that obviously affected the extraction efficiency.

  1. Section 4.3. Information about the UAE device is missing.

Answer: I have included UAE device information in the article in line 471.

  1. HPLC method should be explained briefly.

Answer: I have adjusted the order of the articles and briefly explained the HPLC methods in line 496.

  1. Lines 114-130: Why is this give in Results?

Answer: According to your suggestion, I have reordered the articles.

  1. Experiments were performed in Box-Behnken experimental design which is suitable for RSM. How is that compatible with least squares support vector machine?

Answer: R-squared is to evaluate the prepared prediction models and compare their prediction results with the actual situation. The larger R-squared is, the better the fitting degree is. Mean-square error (MSE) is a measure that reflects the degree of difference between an estimator and an estimator. The mean square error is the general standard of evaluation point estimation, and the smaller the estimated MSE is, the better. In this paper, compared with BBD-RSM, BBD-LSSVM also has better R2 and MSE, indicating that both BBD-LSSVM and BBD-RSM have great applicability.

  1. You compared RSM and least squares support vector machinein terms of optimization. Experiments should be performed at optimal conditions again to check and validate results.

Answer: I have carried out validation experiments under the best conditions of each model, and explained that the extraction results under the conditions of LSSVM model is better than RSM model.

  1. Comparison with literature is also missing. RSM has been compared with other optimization tools in numerous studies recently. It should have that in mind for improvement of discussion.

Answer: I apologize for the errors in the literature. I have read and organized more literature, and refined the discussion according to your opinion.

  1. What about comparison on other levels? What is more suitable for model fitting? What is more suitable for influence analysis? Authors should look beyond the optimization to improve the scientific impact of the paper.

Answer: According to your opinion, I have discussed in detail the influence of different extraction factors on extraction efficiency and the advantages of different models.

  1. Discussion is rather modest and should be reassessed.

Answer: I have renewed the discussion in detail

  1. Conclusions section is utterly problematic. Authors should put some serious effort and write the appropriate conclusions.

Answer: I have rewritten the conclusion and written the appropriate conclusion

  1. Title of the paper is not suitable since antioxidant activity and bioactive compounds were not simultaneously optimized! It should be rewritten.

Answer: I've rewritten it as “Optimization of ultrasonic-assisted extraction of active components from Polygala tenuifolia: A comparative study of the response surface methodology and least squares support vector machine”

Reviewer 2 Report

This paper is an originally and innovative study.

In general, the manuscript is well structured and easy to read. the introduction is concise and gives the reader an idea about the used methods and their behaviors. The materials, results, discussion, and conclusions are clear because it has been divided in several groups which helps the reader to follow the work.  In line 44, please modify “in genal” by “in general”.

Author Response

Thank you very much for your letter and advice. We have revised the manuscript, and would like to re-submit it for your consideration. We have addressed the comments raised by the reviewers, and the amendments are highlighted in red in the revised manuscript. Point by point responses to the reviewers’ comments are listed below this letter.

I have corrected the spelling and other mistakes in the article.

Reviewer 3 Report

The aim of the study was to optimize the ultrasound-assisted extraction conditions for bioactive (antioxidant) compounds from Yuanzhi by Box-Behnken design. The effects of extraction time, temperature, liquid-solids ratio and ethanol concentrations were investigated. Although the ultrasound-assisted extraction method is often used both for scientific purposes and in the food industry, the use of this technique to extract DISS and PolyIII was not the subject of previous research. The innovative nature of this work is the selection of research material that is little known in the world, and research shows that it has a positive impact on human health. On the other hand, the fact that the research material comes from a given region of the world makes the research of regional importance. However, I believe the topic is innovative and increases our understanding of UAE technique.

In the introduction, no information is given about the substances that are the subject of research. We only know their chemical formula and are informed that they have antioxidant properties. While the ultrasound-assisted extraction method is widely used, the Yuanzhi herb is little known in Europe and the world. Also, the significant effect of yuanzhi-derived DISS and PolyIII is little known, except in China. Therefore, I recommend writing more information about DISS and PolyII in the introduction. Their properties are already known (e.g. neuronal cell protective or antidepressant) and what their recommended doses are.

The analytical methods and the experimental design are properly described, but more details are required.

  • Ethanol was used for extraction. What was that ethanol? About what purity and what company it came from. As the use of this extractant is an important aspect of the work, details about it should be provided in the description of the materials, the more so as the origin of ethanol may be different.
  • What type of membrane was used in the purification of HPLC samples?
  • The model and the company of the ultrasonic cleaner used in the research should be given.
  • In the description of the analytical methods, there is no HPLC determination method for the tested compounds. According to the information from the HPLC analysis, the determination of PolyIII and DISS content was made according to the method described in publication 32 (He et al. 2022). In this publication, the content of dioscin and diosgenin in P. kingianum was determined by HPLC. I believe that the entire analytical method for the determination of PolyIII and DISS should be described in the article, and the information on the determination of the antioxidant activity may be shortened, because they are commonly used methods.

The weakest part of the manuscript is the discussion of the obtained results and conclusions. The discussion should be extended to an attempt to explain (taking into account the nature of the extracted compounds and the solvent and the method of extraction) why the parameters influenced the efficiency of extraction. It would also be good to compare the obtained results with those that were carried out and described in previous publications. Currently, this discussion is actually a summary of the description of the results and makes the research performed less relevant.

At work, some drawings are low resolution and it is difficult to read the legend of the drawings or the signatures of the axes of the graphs. The font of these charts should be enlarged (figures 2, 2, 5).

There are also two graphs number 2 in the manuscript. After table number 3 there is table number 1. It all needs to be sorted out.

Figure 3 signature is not accepted. In fact, most charts can be labeled this way. There is also no description of the X and Y axes.

The text of the article has stylistic errors, so I think it should be checked by a native speaker.

Author Response

Thank you very much for your letter and advice. We have revised the manuscript, and would like to re-submit it for your consideration. We have addressed the comments raised by the reviewers, and the amendments are highlighted in red in the revised manuscript. Point by point responses to the reviewers’ comments are listed below this letter.

  1. In the introduction, no information is given about the substances that are the subject of research. We only know their chemical formula and are informed that they have antioxidant properties. While the ultrasound-assisted extraction method is widely used, the Yuanzhi herb is little known in Europe and the world. Also, the significant effect of yuanzhi-derived DISS and PolyIII is little known, except in China. Therefore, I recommend writing more information about DISS and PolyII in the introduction. Their properties are already known (e.g. neuronal cell protective or antidepressant) and what their recommended doses are.

Answer: According to your opinion, I have added more information about Yuanzhi, DISS and PolyIII to the article.

  1. The analytical methods and the experimental design are properly described, but more details are required.

Answer: I have adjusted and improved the experimental methods and design

  1. Ethanol was used for extraction. What was that ethanol? About what purity and what company it came from. As the use of this extractant is an important aspect of the work, details about it should be provided in the description of the materials, the more so as the origin of ethanol may be different.

Answer: Ethanol (The mass fraction of ethanol ≥99.7%, CAS:64-17-5) was purchased from SHUANGLIN chemical reagent Co., LTD. (Hangzhou, China). And I have added information about ethanol to the text

  1. What type of membrane was used in the purification of HPLC samples?

Answer: 0.22micron microporous membrane were purchased from Jingteng Co., LTD (Tianjin, China). This time I have included it in the article.

  1. The model and the company of the ultrasonic cleaner used in the research should be given.

Answer: I have added information about ultrasonic cleaner to the article.

  1. In the description of the analytical methods, there is no HPLC determination method for the tested compounds. According to the information from the HPLC analysis, the determination of PolyIII and DISS content was made according to the method described in publication 32 (He et al. 2022). In this publication, the content of dioscin and diosgenin in P. kingianum was determined by HPLC. I believe that the entire analytical method for the determination of PolyIII and DISS should be described in the article, and the information on the determination of the antioxidant activity may be shortened, because they are commonly used methods.

Answer: According to your opinion, I have made some adjustments to the article and made necessary supplements to THE HPLC part.

  1. The weakest part of the manuscript is the discussion of the obtained results and conclusions. The discussion should be extended to an attempt to explain (taking into account the nature of the extracted compounds and the solvent and the method of extraction) why the parameters influenced the efficiency of extraction. It would also be good to compare the obtained results with those that were carried out and described in previous publications. Currently, this discussion is actually a summary of the description of the results and makes the research performed less relevant.

Answer: Thank you very much for your advice. I have rewritten the discussion according to your opinion. This time I discussed in detail the factors affecting extraction and the relationships between models.

  1. At work, some drawings are low resolution and it is difficult to read the legend of the drawings or the signatures of the axes of the graphs. The font of these charts should be enlarged (figures 2, 2, 5). There are also two graphs number 2 in the manuscript. After table number 3 there is table number 1. It all needs to be sorted out.

Answer: I apologize for the errors in the picture. I have finished proofreading the pictures.

  1. Figure 3 signature is not accepted. In fact, most charts can be labeled this way. There is also no description of the X and Y axes.

Answer: In the figure, the kernel parameter G and penalty factor C are parameters of the model without specific meaning.

Round 2

Reviewer 1 Report

Paper was improved according to my suggestions and remarks.

Reviewer 3 Report

All my comments have been incorporated in the revised manuscript. Incomprehensible statements appeared in the text, eg Finally [陈 9] line 132, sample set [陈 10] line 214, s [陈 13] line 328 and 334. This should be removed from the text or translated into English.